# Prolonged QT Interval in HIV-1 Infected Humanized Mice Treated Chronically with Dolutegravir/Tenofovir Disoproxil Fumarate/Emtricitabine [note 1]

**DOI:** 10.3390/ijms27010519

**Published:** 2026-01-04

**Authors:** Ali Namvaran, Julian V. Garcia, Mahendran Ramasamy, Kayla Nguyen, Farzaneh Tavakkoli Ghazani, Bryan T. Hackfort, Prasanta K. Dash, Reagan E. Fisher, Benson Edagwa, Santhi Gorantla, Keshore R. Bidasee

**Affiliations:** 1Department of Pharmacology and Experimental Neuroscience, University of Nebraska Medical Center, Omaha, NE 68198-5800, USA; anamvaran@unmc.edu (A.N.); juliangarcia@creighton.edu (J.V.G.); mramasamy@unmc.edu (M.R.); kaynguyener@gmail.com (K.N.); ftavakkolighazani@unmc.edu (F.T.G.); pdash@unmc.edu (P.K.D.); refisher@unmc.edu (R.E.F.); benson.edagwa@unmc.edu (B.E.); sgorantla@unmc.edu (S.G.); 2Cellular and Integrative Physiology, University of Nebraska Medical Center, Omaha, NE 68198-5800, USA; byran.hackfort@unmc.edu

**Keywords:** HIV-1, humanized mice, electrocardiography, sudden cardiac death, antiretroviral therapy, arrhythmia, QT interval

## Abstract

The REPRIEVE Trial recently reported high rates of sudden cardiac death (SCD) middle-aged people living with HIV-1 infection (PWH) using the WHO/NIH-recommended two nucleoside reverse transcriptase inhibitors (NRTIs)/one integrase strand inhibitor (INSTI) regimen to manage HIV-1 viremia. To date, clinically relevant animal models to delineate underlying causes for this remain limited. Here, we assessed if HIV-1-infected NOD.Cg-*Prkdc^scid^Il2rg^tm1Wjl^*/SzJ humanized mice (Hu-mice) treated with the WHO/NIH-recommended antiretroviral regimen, dolutegravir (DTG, INSTI)/tenofovir disoproxil fumarate (TDF, NRTIs)/emtricitabine (FTC, NRTIs), can recapitulate abnormalities in the ECG and subclinical structural heart disease that serve as harbingers of SCD in middle-aged PWH. HIV-1-infected and uninfected Hu-mice served as controls. After one month of infection (HIV-1_ADA_), ECG intervals/segments were significantly altered. ECG changes progressively worsened as the duration of untreated infection increased. Treating HIV-1-infected animals with the DTG/TDF/FTC for eight weeks, starting four weeks after infection, prevented worsening, but did not restore ECG intervals/segments to those before infection. In hearts from DTG/TDF/FTC-treated animals, steady-state levels of the sarco-(endo) plasmic reticulum Ca^2+^ ATPase (SERCA2) were reduced by 35%. Steady-state levels of type 2 ryanodine receptor (RyR2) did not change, but its phosphorylation status at Ser2808 was 2-fold higher than that of uninfected controls, indicative of a gain-of-function. The density of perfused micro vessels and fibrosis in hearts of DTG/TDF/FTC-treated animals was not significantly different from that of HIV-1-infected and uninfected Hu-mice. These data show for the first time that HIV-1 infection is triggering abnormalities in the ECG of Hu-mice, and changes in ECG persisted with DTG/TDF/FTC treatment, independent of ischemia and/or fibrosis. They also indicate that chronic DTG/TDF/FTC treatment did not worsen ECG changes, including the QT interval. Since phosphorylation of RyR2 at Ser2808 occurs via β-adrenergic activation of protein kinase A, these new data also suggest that chronic hyperadrenergic activity may be increasing the risk of SCD via Ca^2+^ leak through RyR2.

## 1. Introduction

Combination antiretroviral therapy has increased life expectancy for the majority of the ~40 million individuals with HIV-1 infection (PWH) [1,2]. However, in many of these individuals, early-onset cardiovascular diseases continue to be a major medical challenge that negatively impacts their quality of life [3,4,5]. The rate of sudden cardiac death (SCD) is also >4-fold higher in PWH [6,7,8,9,10,11,12], with a higher incidence in males than in females [11,13,14]. Studies suggest that the increased rate of SCD in PWH is linked to abnormalities in the electrocardiogram (ECG), including an increase in the PR interval, changes in the P-wave amplitude and duration, prolongation of the QT interval, widening of the QRS complex, and an increase in the ST-T segment [15,16,17,18,19,20]. Of greatest clinical concern is the prolongation of the QT interval that serves as a harbinger for ventricular tachyarrhythmia, cardiac arrest, and SCD, especially in out-of-hospital settings [21].

To date, studies suggest that the increase in the QT interval in PWH arises principally from downregulation and/or blockage of transient outward K+ currents (I_to_) and delayed rectifying K+ currents (IKr, KCNH2/hERG, and IKs, KCNE1) by HIV-1 proteins, select (older) antiretroviral drugs, and approved and illicit drugs [16,17,18,19,22]. Others suggest that HIV-associated neuropathy, hepatic and renal insufficiencies, ischemia, fibrosis, tissue remodeling (structural heart disease), electrolyte imbalance, and metabolic alterations are also contributing to the prolongation of the QT interval [20,23,24,25,26,27,28,29]. However, it is not clear whether the latter are initiating causes or exacerbating/worsening early-onset ECG changes [30].

Over the last two decades, various combinations of antiretroviral drugs have been used to lower HIV viremia in antiretroviral naïve patients, children/adolescents, and women of childbearing age and to reduce cardiovascular disease risks, including SCD in PWH. In 2019, the WHO [31], NIH [32], and the British HIV Association [33] recommended using an integrase strand inhibitor (INSTI) with two nucleoside reverse transcriptase inhibitors (NRTIs) as a first line regimen for antiretroviral-naïve patients. Three INSTIs are currently in use to lower plasma HIV-1 viremia in antiretroviral naïve patients: raltegravir (RAL), dolutegravir (DTG), and bictegravir (BIC). Tenofovir alafenamide (TAF), tenofovir disoproxil fumarate (TDF), emtricitabine (FTC), lamivudine (3TC), and abacavir (ABC) are widely used NRTIs [32]. The two NRTIs/one INSTI combination was selected in part because of the high potencies of these drugs and their limited effects at supratherapeutic doses on the QT interval of the ECG in acute studies [12,34,35,36,37,38,39,40].

Very recently, deFilippi et al. [12] reported a high incidence of SCD in the REPRIEVE Trial participants that exhibited low to intermediate 10-year risk of developing atherosclerotic cardiovascular disease and were using two NRTIs/one INSTI to maintain low plasma HIV-1 viremia. Of the twenty-five patients that died over the five years, nine (36%) were on the two NRTIs/one INSTI regimen. From the two NRTIs/NNRTI regimen, eight participants died from SCD (32%) and from the two NRTIs/one protease inhibitor group, three participants died from SCD (12%). These new data suggest that contrary to acute studies, chronic usage of NRTIs and INSTI are also likely prolonging the QT interval of the ECG. What is not yet clear is whether mechanisms other than delaying cardiac repolarization such as a reduction in Na+ current [21,41], alterations in the activity of cardiac L-type Ca^2+^ current [42,43], and aberrant sarcoplasmic reticulum (SR) Ca^2+^ release and uptake [44,45,46,47,48,49,50,51] are also contributing to the increased risk of ventricular tachyarrhythmia and cardiac arrest.

Non-human primates, transgenic rodents, EcoHIV mice, and humanized mice are available models to delineate mechanisms that contribute to end-organ complications, including heart failure, SCD, HIV reservoirs, and to search for HIV-1 cures [52,53,54,55]. Despite significant contributions of non-human primate models for HIV-1 therapeutic design and vaccine research, SIV infections do not exactly recapitulate HIV-1 infections [56,57]. Moreover, non-human primates are expensive to house and maintain. While transgenic rodent models can provide insights into the mechanism, they usually express higher levels of HIV-1 auxiliary proteins than those reported in PWH, do not adequately reflect the disease state, and usually do not respond to treatment with antiretroviral drugs. In previous studies, Fiset and colleagues reported prolonged cardiac repolarization and reduced Na+ currents in CD4C/HIV transgenic mice [24,41]. The EcoHIV mouse, which is made by replacing the HIV-1 gp120 gene with the ectopic MLV gp80 gene for entry into cells, is an adequate model for studying end-organ complications in HIV-1 but lacks some of the characteristics of HIV-1 infection. To the best of our knowledge, there is also a paucity of information on ECG changes in EcoHIV mice and SIV-infected non-human primates.

Earlier, we showed that NOD.Cg-*Prkdc^scid^Il2rg^tm1Wjl^*/SzJ humanized mice (Hu-mice) infected with HIV-1 develop a progressive heart failure similar to that in PWH [58], and treating Hu-mice (female) with DTG/TDF/FTC for 12 weeks starting 4 weeks after infection did not blunt diastolic dysfunction [59]. Since the first one-third of cardiac relaxation is dictated by the activity of sarco(endo)plasmic reticulum Ca^2+^-ATPase (SERCA2), the protein responsible for pumping Ca^2+^ back into the SR, these data suggest that SR Ca^2+^ cycling may be perturbed in HIV-1-infected Hu-mice with and without antiretroviral drug treatment increasing the risk of ventricular tachyarrhythmia. Therefore, the purpose of this study was three-fold: (i) to determine if HIV-1-infected Hu-mice treated chronically with DTG/TDF/FTC for eight weeks, starting four weeks after infection also develop changes in ECG including prolongation in QT interval akin to that reported in PWH; (ii) to determine if steady state levels/function of the primary SR Ca^2+^ cycling proteins, SERCA2 and ryanodine receptor Ca^2+^-release channel (RyR2) are changed in hearts of HIV-1 infected Hu-mice treated with DTG/TDF/FTC for eight weeks, starting four weeks after infection; and (iii) to assess the density of perfused micro vessels/ischemia and fibrosis in Hu-mice infected treated with DTG/TDF/FTC for eight weeks as indices of structural heart disease. Uninfected and HIV-1-infected Hu-mice serve as controls.

## 2. Results

### 2.1. Characteristics of Animals Used in the Study

Longitudinal changes in body weight (grams) of animals used in this study are shown in Table 1. At the start of the study, human CD45+ immune cell reconstitution in the 24 Hu-mice ranged from 20 to 40% of peripheral blood white blood cells. HIV-1 infection resulted in productive infection, with plasma HIV-1 viral load reaching 3.0 ± 0.2 × 10^5^ RNA copies/mL after 4 weeks of infection and persisting at this level for 12 weeks post-infection (2.9 ± 0.2 × 10^5^ RNA copies/mL) (Figure 1A). After 4 weeks of treatment with DTG/TDF/FTC, plasma viral loads in HIV-1-infected mice decreased to 2.0 ± 0.2 × 10^3^ RNA copies/mL, and after 8 weeks of treatment, plasma viral loads decreased further to 205 copies/mL (Figure 1A). The percentages of CD4+ T cells in blood also progressively declined in HIV-1-infected mice from 66.8 ± 0.4% to 39.2 ± 2.1% over the 12-week protocol, while that of CD8+ T cells gradually increased over the same period from 28.8 ± 1.3% to 52.2 ± 6.1% (Figure 1B,C). Treatment with DTG/TDF/FTC for eight weeks restored CD4+ and CD8+ T cells to levels near that of uninfected controls (Figure 1B,C).

### 2.2. ECG Traces

ECG parameters measured included P wave amplitude/duration, PR segment, PR interval, QRS complex duration, QT interval, ST segment, ST-T segment, TP interval, TQ interval, RR and PP intervals, R wave amplitude, S wave amplitude, and J wave amplitude, over at least four consecutive cardiac cycles (not during respiration) (Figure 2). All interval measurements were corrected for heart rate using Bazett’s formula: QTc = QT/(RR/100)^1/2^ [60,61,62]. Figure 3 shows representative Lead II ECG traces from a male uninfected control (green), HIV-1-infected (red), and HIV-1-infected treated mice treated with DTG/TDF/FTC (blue) at the end of the study protocol (12 weeks). Similar ECG patterns were also observed in female mice, and data from each group were pooled. There were no significant changes (*p* > 0.05) in the R-R intervals (and P-P interval) between uninfected controls, and HIV-1-infected with and without DTG/TDF/FTC treatment. However, the other segments of the ECG from HIV-1 infected with and without DTG/TDF/FTC treatment were noticeably different from those from uninfected animals. First, in HIV-1 infected animals, the amplitude and duration of the P-wave, the PRc interval, and P-R segment were increased (*p* < 0.05, Figure 4). Second, the QRS complex, S-wave amplitude, J-wave amplitude, and QTc segment were also increased (*p* < 0.05, Figure 5). Finally, STc and ST-Tc intervals were significantly increased (*p* < 0.05), while TPc and the TQc intervals were decreased (*p* < 0.05, Figure 6). These changes started as early as 4 weeks after infection and progressively worsened as the duration of infection increased (Figure 4, Figure 5 and Figure 6). Treating HIV-1-infected Hu-mice with the DTG/TDF/FTC for eight weeks, starting four weeks after infection, prevented ECG from worsening, but did not revert to pre-infection (or uninfected) state.

### 2.3. Expression of RyR2, phosphoRyR2 (Ser 2808), and SERCA2

After 12 weeks of infection, RyR2 protein levels remained unchanged (*p* > 0.05). However, phosphorylation at Ser2808 (a protein kinase A site) increased 3-fold (Figure 7A,B). Treating HIV-1-infected animals with DTG/TDF/FTC for eight weeks, starting four weeks post-infection, also did not alter the expression of RyR2, but its phosphorylation at Ser2808 remained 2-fold higher than that of uninfected controls (Figure 7A). After 12 weeks of infection, SERCA2 protein levels were 35% lower than those in hearts from uninfected controls (Figure 7A,C). Treating infected animals with antiretroviral drugs did not restore SERCA2 to pre-infection levels (Figure 7A,C).

### 2.4. Density of Perfused Micro Vessels, Microvascular Leakage, Ischemia, and Fibrosis

The density of perfused micro vessels in the hearts of HIV-1-infected Hu-mice was not significantly different from that of uninfected controls and HIV-1-infected mice treated with DTG/TDF/FTC for eight weeks starting four weeks after infection (Figure 8A,B, *p* > 0.05). There were also no microvascular leakages or ischemia. There were also minimal levels of interstitial and perivascular fibrosis in hearts of HIV-1-infected Hu-mice, uninfected controls, and HIV-1-infected mice treated with DTG/TDF/FTC for eight weeks starting four weeks after infection (Figure 8C–E).

## 3. Discussion

Middle-aged PWH have a >4-fold higher incidence of SCD compared to uninfected individuals of similar age [6,7,8,10,12]. Studies have attributed this to a fatal ventricular tachyarrhythmia from prolongation in the QT interval that arises principally from downregulation and/or blockage of transient outward K+ currents (I_to_) and delayed rectifying K+ currents (IKr, KCNH2/hERG, and IKs, KCNE1) by HIV-1 proteins, select antiretroviral drugs (NNRTIs like efavirenz, and protease inhibitors like atazanavir and ritonavir) [63,64,65,66], and approved and illicit/occult drugs [16,17,18,19,22]. Others also suggest that HIV-associated neuropathy, hepatic and renal insufficiencies, ischemia, fibrosis, tissue remodeling, electrolyte imbalance, and metabolic alterations may also exacerbate the prolongation in the QT interval [20,23,24,25,26,27,28]. deFilippi et al., [12] also reported a high incidence of SCD in PWH on the two NRTIs’/INSTI regimen. The latter was unexpected since acute studies have found that NRTIs’ and INSTIs have little or no effect, even at supratherapeutic doses, on the QT interval of the ECG [12,34,35,36,37,38]. To date, clinically relevant animal models to investigate whether chronic NRTIs and INSTI usage are negatively impact ECG, including QT interval in the setting of HIV-1 infection remain undefined.

The principal finding of the present study is that NOD.Cg-*Prkdc^scid^Il2rg^tm1Wjl^*/SzJ humanized mice (Hu-mice) of both sexes exhibited multiple and progressive changes in their ECG starting four weeks after infection. These changes include increases in the amplitude and duration of the P-wave, PR interval, PR segment, QRS complex, S-wave amplitude, J-wave amplitude, QT interval, ST segment, and ST-T segment, along with decreases in TP and TQ (see Figure 3). The PR interval of the ECG is a measure of the time taken for the electrical impulse to travel from the atria to the ventricles, reflecting atrioventricular conduction time. An increase in the PR interval is suggestive of right atrial enlargement or dysfunction and can serve as a substrate for supraventricular arrhythmias (atrial fibrillation, AF), a clinical feature seen in PWH [67,68,69]. At high heart rates, AF can progress to ventricular arrhythmias [16]. The QRS complex reflects ventricular depolarization/start of contraction. The T wave represents the time to ventricular repolarization, i.e., return of the ventricles to their resting electrical state. The QT interval represents the time taken for ventricular depolarization/repolarization. Prolongation in ST-segment and T-wave reflects a slowed repolarization, while decreases in TP and TQ intervals reflect shortening in time of cardiac electrical silence (relaxation and time available for filling). These new data indicate that HIV-1 infection is sufficient to alter the ECG of Hu-mice, and these changes start as early as four weeks after infection and in the absence of antiretroviral drug treatment.

Another major finding of this study is that treating HIV-1-infected Hu-mice with DTG/TDF/FTC for eight weeks, starting four weeks after infection, prevented worsening but did not restore the ECG pattern to that seen pre-infection. This combination of antiretroviral drugs (DTG/TDF/FTC) was approved for use in 2019 and is not known to directly impact the QT interval [34,35,37,38,39,40] like some of the “older antiretroviral agents” [64,65,66,70]. However, in DTG/TDF/FTC-treated animals, the P-wave amplitude and duration remained increased, the PR interval remained increased, and the QT interval (widening of the QRS complex and increases in the T-wave duration and ST-T interval) remained increased [23,24]. From these new data, we posit that intrinsic factors, including residual HIV-1 proteins, sympathetic activation, oxidative stress, inflammation, and metabolic alterations are driving the early-onset ECG changes, and these changes persist with DTG/TDF/FTC-treatment. This model is well-positioned to delineate which illicit and approved drugs are triggering ventricular tachyarrhythmia and cardiac arrest, as well as identify pharmacological strategies to blunt prolongation in the QT interval and ventricular tachyarrhythmias. At present, aggressive control of HIV viremia, cardiac risk factors, and abstinence from unhealthy behaviors remain pillars to prevent heart failure and ventricular tachyarrhythmia in PWH [23,24,71,72]. Alvi et al. [10] reported earlier that in older people with heart failure and HIV (PWH), beta-blocker usage attenuated hyperadrenergic function to decrease the risk of sudden cardiac death (SCD).

Aberrant release of Ca^2+^ from the SR and a slowing in the uptake of Ca^2+^ into the SR can also serve as triggers for ventricular tachyarrhythmias [44,45,46,47,48,50,51,73]. However, little is known about whether expression and/or activities of the major SR Ca^2+^ release and uptake proteins, namely RyR2 and SERCA2 are altered during HIV-1 infection, and whether these changes are prevented/reversed with antiretroviral drug treatment. Another major finding of the present study is that phosphorylation of RyR2 at Ser2808 was 3-fold and 2-fold higher in hearts of HIV-1-infected and HIV-1-infected Hu-mice treated with DTG/TDF/FTC, indicative of a gain-of-function. Phosphorylation of RyR2 at Ser2808 arises from protein kinase A (PKA) activation following stimulation of β-adrenergic receptors (β-AR) [74]. Studies have also reported increased sympathetic activity during early HIV-1 infection [75,76]. The increased inflammation and oxidative stress reported in PWH may be contributing to the sympathetic excitation [77]. The norepinephrine release during sympathetic excitation will stimulate β-adrenergic receptors (β-AR) [74].

Ser2030 is another PKA phosphorylation site on RyR2. Persistent phosphorylation of RyR2 at Ser2808 and Ser2030 will result in a gain-of-function and Ca^2+^ leak from the SR during diastole [47]. Leakage of Ca^2+^ from the SR during diastole is associated with arrhythmogenic disorders. The arrhythmogenic mechanism arises from delayed afterdepolarizations (DADs), triggering premature ventricular contractions [11,50,78,79,80,81]. Persistent phosphorylation of RyR2 will also increase the propensity for spontaneous Ca^2+^ releases, another mechanism that could contribute to higher susceptibility to ventricular arrhythmias, cardiac arrest and SCD, especially in out-of-hospital setting. More work is needed to determine if expression of proteins that bind to and modulate the activity of RyR2, such as calsequestrin and triadin [78] are also being altered in the hearts of HIV-1-infected Hu-mice without and with DTG/TDF/FTC treatment.

There is a third phosphorylation site on RyR2, i.e., Ser2814, and this site is phosphorylated by Ca^2+^/calmodulin kinase II. Ca^2+^/Calmodulin kinase II belongs to a family of enzymes that are activated by increases in cytoplasmic Ca^2+^ [82]. CaMKII contains a N-terminal catalytic domain, a C-terminal association domain, and a regulatory domain. The regulatory domain contains the autoinhibitory Ca^2+^/calmodulin binding region. When cellular Ca^2+^ increases, Ca^2+^ binds to calmodulin, will remove autoinhibition, and the active enzyme is free to phosphorylate cellular proteins, including RyR2. Aberrant activation of RyR2 due to increased phosphorylation at Ser2808 and Ser2030 will elevate cytoplasmic Ca^2+^, activate Ca^2+^/calmodulin kinase, which in turn will phosphorylate RyR2 at Ser2814, further increasing the propensity of Ca^2+^ leak during diastole, ventricular tachyarrhythmia, cardiac arrest, and SCD. These data support those of Alvi et al. [10] who earlier reported that β-blockers by reducing hyperadrenergic function decrease the risk of sudden cardiac death (SCD) in older people with heart failure and HIV (PWH).

In this study, we also found a 35% reduction in SERCA2 in hearts from HIV-1-infected Hu-mice with and without DTG/TDF/FTC treatment. Cardiac relaxation occurs in two stages. The first stage arises when SERCA2 returns the Ca^2+^ released from the SR, and the second stage depends on the distensibility of the extracellular matrix of the heart (i.e., fibrosis) [83,84]. When the activity of SERCA2 is slowed due to a reduction in expression, the time to cardiac relaxation is slowed, and a sudden tachycardia can elicit ventricular arrhythmias [46,51]. Expression of SERCA2 is regulated by thyroid hormones (T3 and T4), and studies have reported hypothyroidism in PWH [85,86,87]. Williams et al. recently reported that HIF-1α-dependent upregulation of miR-29c can also inhibit SERCA2 expression and reduce cardiac contractility [88]. Bekeredjian et al. [89] also showed that overexpression of HIF-1α—downregulates SERCA2. Recently, we showed that increased expression of HIF-1α in hearts of HIV-1-infected DTG/TDF/FTC-treated Hu-mice and in H9C2 cardiac myocytes exposed to DTG, TDF, and FTC in ambient oxygen (20%) [59]. Studies are also ongoing to determine if expression of proteins that modulate the activity of SERCA2, including phospholamban and sarcolipin, are also changing in hearts of HIV-1-infected Hu-mice with and without DTG/TDF/FTC treatment.

Another major finding of the present study is that the density of perfused micro vessels in the hearts of Hu-mice was not significantly altered after 12 weeks of infection. Treating HIV-1-infected mice with DTG/TDF/FTC for eight weeks, starting four weeks after infection also did not alter the density of perfused micro vessels. There was also no evidence of microvascular leakage and ischemia in the hearts of uninfected controls or HIV-1-infected mice treated with DTG/TDF/FTC for eight weeks, starting four weeks after infection. Interstitial and perivascular fibrosis were also minimal in hearts of HIV-1-infected Hu-mice, uninfected controls, and HIV-1-infected mice treated with DTG/TDF/FTC for eight weeks starting four weeks after infection. These new data show that changes in the ECG seen in HIV-1-infected mice treated with DTG/TDF/FTC were not due to subclinical structural heart disease. However, it should be mentioned that we previously reported microvascular leakage, ischemia, and fibrosis in hearts from Hu-mice after 16–17 weeks of infection, indicative of structural heart disease [58]. Since ECG changes were observed as early as four weeks after infection, structural heart disease is not likely to be the initiating cause for alterations in the ECG, including prolongation in the QT interval.

In conclusion, the present study shows that HIV-1 infection is inducing early-onset ECG changes in Hu-mice. Treating HIV-1-infected Hu-mice treated with DTG/TDF/FTC for 8 weeks starting four weeks prevented worsening but did restore the ECG changes to that pre-infection. These data suggest that chronic DTG/TDF/FTC treatment is unlikely to be negatively impacting the QT interval. However, the ECG changes in HIV-1-infected Hu-mice treated with DTG/TDF/FTC were similar to that reported in PWH, including clinically concerning prolongation in the QT interval, but not structural heart disease. The model also shows for the first time phosphorylation of RyR2 at Ser2808 is increased in hearts from HIV-1-infected and HIV-1-infected animals treated with DTG/TDF/FTC, suggesting aberrant release of Ca^2+^ from the SR during diastole could be increasing the risk of ventricular arrhythmias and cardiac arrest. Since an increase in RyR2 phosphorylation at Ser2808 occurs via protein kinase A-dependent activation of β-adrenergic receptors, we posit that β-blockers may also be useful not only in older HIV-infected individuals with heart failure but also in younger PWH to blunt hyperadrenergic activity, attenuate the PKA-dependent gain-of-function of RyR2, prolongation in the QT interval and risk for ventricular tachyarrhythmia. The present study also shows for the first time that expression of SERCA2 is reduced in hearts of HIV-1-infected Hu-mice, and that this reduction was not reversed with DTG/TDF/FTC treatment. However, the mechanisms by which SERCA2 becomes downregulated remain undefined.

This study is not without limitations. First, murine cardiac electrophysiology differs from that of humans in several aspects, including heart rate, action potential duration, and the expression of ion channels involved in ECG. These differences must be considered when extrapolating our findings to PWH. Second, our ECG recordings were performed under isoflurane anesthesia. Isoflurane anesthesia has been shown to independently prolong the QT interval by modulating cardiac ion channel activity and suppressing autonomic tone in mice [90,91]. However, since all animals were subjected to the same anesthetic conditions, we reason that ECG changes are due to HIV-1 infection without and with antiretroviral drug treatment rather than due to isoflurane anesthesia. In future studies we will implant telemetric electrodes to assess ECG recordings in freely moving conscious animals.

## 4. Materials and Methods

### 4.1. Ethical Considerations

Animals used in this study were approved by the University of Nebraska Medical Center (UNMC) Institutional Animal Care and Use Committee (IACUC) #21-100-06 FC, reapproved on 2 December 2025 (heart and pulmonary deficits in HIV-1 infection and IBC #23-07-17 (End Organ deficits in HIV-1 infection and Lupus Nephritis), reapproved 28 July 2023. All animal work was conducted in compliance with UNMC institutional policies and the NIH Guide for Use and Care of Laboratory Animals. Animals were housed in a pathogen-free barrier facility under controlled environmental conditions, including a 12 h light/dark cycle, an ambient temperature of 22 ± 2 °C, 40–60% humidity, and free access to food and water.

### 4.2. Antibodies and Reagents

Antibodies to RyR2 (rabbit polyclonal, PA5-104444), phospho-RyR2 at Ser2808 (rabbit polyclonal, Cat # PA5-36758), and SERCA2 (mouse monoclonal, Cat # MA3-919) were obtained from Thermo-Fisher Scientific (Grand Island, NY, USA). Antibodies to beta-actin (mouse monoclonal, Cat# sc-47778) were from Santa Cruz Biotechnology, Santa Clara, CA, USA. Secondary antibodies were HRP-conjugated anti-mouse IgG (Cat# 7076S, Cell Signaling Technology, Danvers, MA, USA) and donkey anti-rabbit IgG (Cat # 31458, Thermo-Fisher Scientific, Grand Island, NY, USA). Bovine serum albumin labeled with fluorescein isothiocyanate (BSA-FITC, Cat# A9771) and Trichrome (Masson) staining kit (Cat# HT15-1KT) were from Sigma-Aldrich (St Louis, MO, USA). Normal chow (protein 25%, carbohydrate 58%, fat 17%) was from Teklab, Research Diets, New Brunswick, NJ, USA. DietGel^®^ Boost feed (for adding ARDs) was obtained from ClearH2O (Westbrook, ME, USA). Dolutegravir (DTG) and tenofovir disoproxil fumarate (TDF) were provided by Dr. Benson Edagwa (University of Nebraska Medical Center, Omaha, NE, USA). HIV-1_ADA_ to infect mice was provided by UNMC HIV-1 Virus Core. All other reagents used were of the highest-grade quality commercially available.

### 4.3. Generation of NSG Humanized Mice

Humanized mice (Hu-mice, n = 24) used in this study were generated using previously described protocols [58,92]. For this study, animals with CD45+ cells ranging between 20% and 40% were used. During humanization, mice were given normal chow (protein 25%, carbohydrate 58%, fat 17%, Teklab Research Diets, New Brunswick, NJ, USA) with ad libitum access to food and water.

### 4.4. Study Protocol

After humanization (20 weeks after intrahepatic injection of CD34^+^ human hematopoietic stem cells isolated from umbilical cord blood), feed was switched from normal chow to DietGel^®^ Boost gel for one week for acclimatization. After this, Hu-mice (n = 24, 12 males and 12 females) were lightly anesthetized with isoflurane (1–2%, Cardinal Health, Dublin, OH, USA) and placed on a temperature-controlled plate maintained at 37 ± 2 °C. Lead II ECG signals were recorded using a Fujifilm Visual Sonics Vevo 3100 system and accompanying THM150 Physiology Monitoring System (Fujifilm Visual Sonics, Toronto, ON, Canada). All measurements were collected between 9:00 am and 1:00 pm. ECG waveforms collected were analyzed by a blind investigator using FUJIFILM Visual Sonics offline software Vevo LAB 5.7.1.

The next day, Hu-mice were randomly assigned to two groups labeled Group I and Group II of eight and sixteen mice, respectively. Mice in Group II were then infected intraperitoneally with 2 × 10^4^ tissue culture infectious dose 50 (TCID_50_) of HIV-1_ADA_. Group I mice were injected intraperitoneally with the same volume of saline to serve as age-matched uninfected controls (uninfected). Peripheral blood was collected after four weeks of infection via the submandibular vein to assess HIV-1 viral RNA from Group II animals. Plasma HIV-1 RNA levels were measured using an automated COBAS Ampliprep V2.0/Taqman-48 system (Roche Molecular Diagnostics, Basel, Switzerland) as per the manufacturer’s instructions.

### 4.5. Antiretroviral Treatment

Four weeks post-infection, Group II animals were randomly divided into two subgroups, Group IIA and IIB of eight animals each (four males and four females). For animals in Group IIB, dolutegravir (DTG), tenofovir disoproxil fumarate (TDF), and emtricitabine (FTC) were mixed into the DietGel^®^ Boost gel to afford final concentrations of 48 mg/kg/day DTG, 60 mg/kg/day TDF, and 60 mg/kg/day FTC based on 4 g/day consumption [93]. DietGel^®^ Boost gel with DTG/TDF/FTC was refreshed every three days, and treatment continued for a period of 8 weeks. Peripheral blood was collected every four weeks from Group IIA and Group IIB animals to assess HIV-1 viral RNA. ECG was repeated at four and 12 weeks.

### 4.6. Western Blotting Analysis

In a Biosafety Level 3 room, approximately 20 mg of left ventricular tissues from uninfected, HIV-1-infected, and HIV-1-infected animals treated with DTG/TDF/FTC (n = 4–8 per group) were placed into pre-labeled Eppendorf tubes. Two hundred microliters of lysis buffer (200 μL, Thermo-Fisher Scientific, Grand Island, NY, USA) were incubated on ice for 20 min. The tissues were transferred onto a glass slide, cut into small pieces, and returned to the lysis buffer for an additional 10 min. Samples were sonicated 12 times for 5 s at 50% amplitude, with cooling on ice for 10 s between pulses to prevent overheating. After sonication, the samples were centrifuged at 1000× *g* for 5 min at 4 °C, and supernatants were carefully collected and transferred into a new tube kept on ice. The pellets were resuspended in 100 μL of lysis buffer, sonicated 5 times for 5 s each as above, and centrifuged at 1000× *g* for 5 min at 4 °C. The resulting supernatants were combined with the first collection and centrifuged at 2000× *g* for 5 min at 4 °C. Protein concentrations were then measured using the BCA assay (Pierce™ BCA Protein Assay Kits, Cat # A65453, Thermo-Fisher, Grand Island, NY, USA).

Thirty micrograms (30 μg) of protein from each of the three samples were added to gel dissociation medium containing 10 mg dithiothreitol/mL and boiled for 10 min. Samples were then loaded onto 4–20% Tris-Glycine Plus WedgeWell™ gels and separated by SDS-PAGE (2 h, 150 mV). After separation, samples were transferred to polyvinylidene fluoride (PVDF) membranes (Millipore, Billerica, MA, USA) for 2 h at 100 V. Membranes were blocked in 5% non-fat dry milk prepared in 1x TBST buffer (20 mM Tris-HCl, pH 7.6; 150 mM NaCl; 0.1% Tween-20) for 1 h at room temperature and washed three times X10 min with TBST. Membranes were then probed for type 2 ryanodine receptor Ca^2+^ release channel (RyR2), phospho-RyR2 (Ser2808), and sarco(endo)plasmic reticulum Ca^2+^ ATPase (SERCA2). Primary antibody incubation was for 16 h at 4 °C on an orbital shaker with at 1:1000 dilution in 1x TBST. Beta-actin served as internal reference. Incubation with secondary antibodies was for 1 h at room temperature at 1:3000 dilution. Protein bands were visualized using SuperSignal™ West Femto Maximum Sensitivity Substrate (Cat #34095, Thermo-Fisher, Grand Island, NY, USA).

### 4.7. Density of Perfused Micro Vessels and Fibrosis

Starting two days after the last ECG measurement, animals in Group I (uninfected) were anesthetized with 3% isoflurane, and fluorescein isothiocyanate-labeled bovine serum albumin (FITC-BSA 60 mg/kg in sterile 1x PBS buffer, 50 μL) was injected via a tail vein and allowed to circulate for 5 min. After this, the chest cavities were opened, and the hearts were cut in half longitudinally. One-half of each heart was immersed in 4% paraformaldehyde for 36 h at 4 °C. The other half of the hearts were quickly frozen in dry ice and stored at −80 °C until used for biochemical assays. Hearts from HIV-1-infected and HIV-1-infected animals treated with DTG/TDF/FTC were harvested in a comparable way over the next two days.

Paraformaldehyde-fixed hearts were then processed, embedded in paraffin, and 10 μm sections were cut and mounted on glass slides. Sections were then de-paraffinized with xylene (3 changes, ten minutes each) and rehydrated in decreasing concentrations of ethanol (100, 95, 70% and distilled water, three minutes each) followed by a saline wash. Deparaffinized and rehydrated slides were either cover-slipped with Prolong Gold Anti-fade reagent containing DAPI or stained with Masson-Trichrome for fibrosis. Tissues were scanned using a Zeiss Axioscan 7 Whole Slide Imaging System (Zeiss, Hebron, KY, USA) to determine FITC-BSA and Masson-Trichrome staining. Data were quantified using Zen Light Software (Version 3.11.105.044000, Zeiss, Hebron, KY, USA).

### 4.8. Statistical Analysis

Statistical analyses were performed using GraphPad Prism software (version 10.6.1, GraphPad Software, San Diego, CA, USA). Data presented are mean ± standard error of the mean (SEM). Comparisons between the experimental groups at each time point and within groups over time were conducted using one-way analysis of variance (ANOVA) followed by Bonferroni post hoc tests to determine statistical significance. A *p*-value of <0.05 was considered statistically significant.

## Figures and Tables

**Figure 1 ijms-27-00519-f001:**
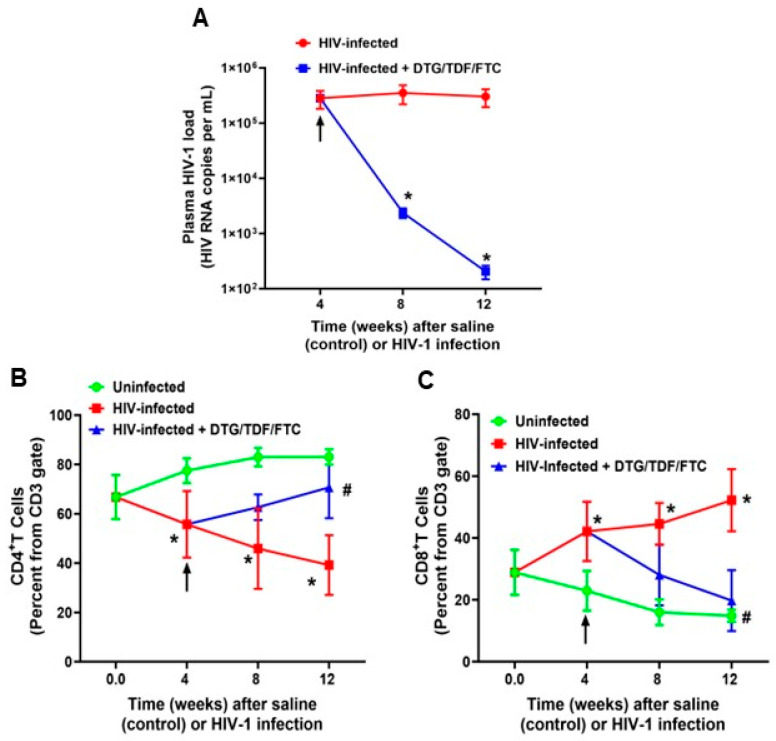
Plasma HIV-1 load and immune cell dynamics in study groups. Panel (**A**) shows plasma HIV-1 load (HIV RNA copies per mL) over time (weeks). Panels (**B**,**C**) show longitudinal changes in CD4+ and CD8+ T cells over time (weeks). Data shown are mean ± SEM, with n = 8 mice per group, four males and four females. All measurements were collected using the LSR-II FACS analyzer (BD Biosciences, Mountain View, CA, USA). Black arrows indicate the start of DTG/TDF/FTC treatment. * Denotes significantly different from uninfected (control), *p* < 0.05; # denotes significantly different from HIV-1-infected, *p* < 0.05.

**Figure 2 ijms-27-00519-f002:**
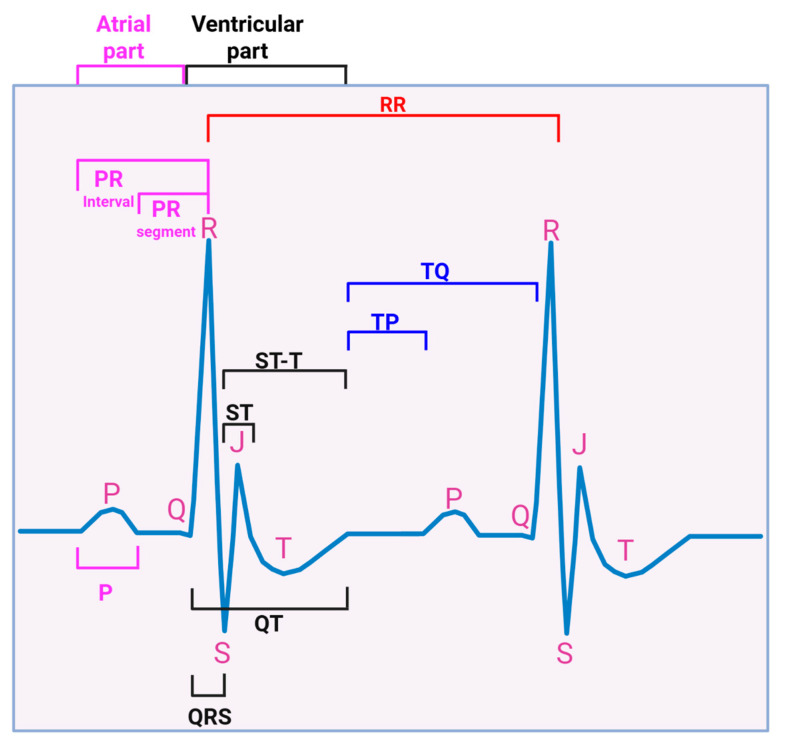
Schematic of a typical mouse electrocardiogram (ECG) showing the various parameters measured in this study. Signals through atria are shown in fuchsia and ventricular signals are in black and blue.

**Figure 3 ijms-27-00519-f003:**
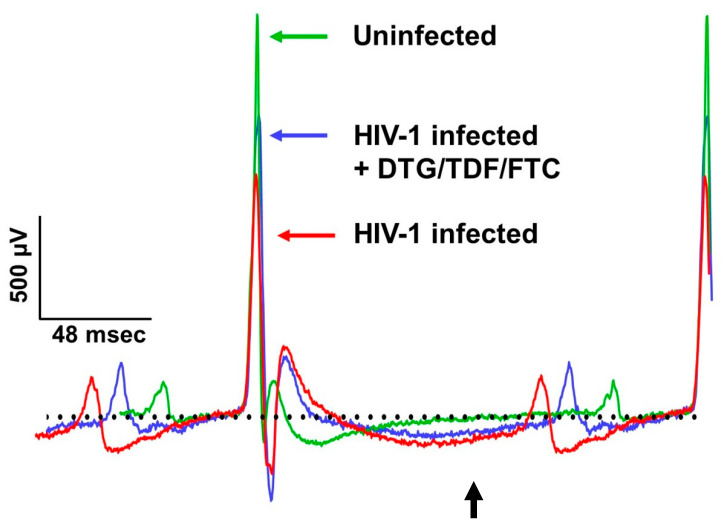
Representative ECG at 12 weeks from male uninfected controls (green), HIV-1-infected (red), and HIV-1-infected Hu-mice treated with DTG/TDF/FTC (blue). Similar changes were observed in female mice. Black arrow shows a prolonged T wave. Black dots represent baseline.

**Figure 4 ijms-27-00519-f004:**
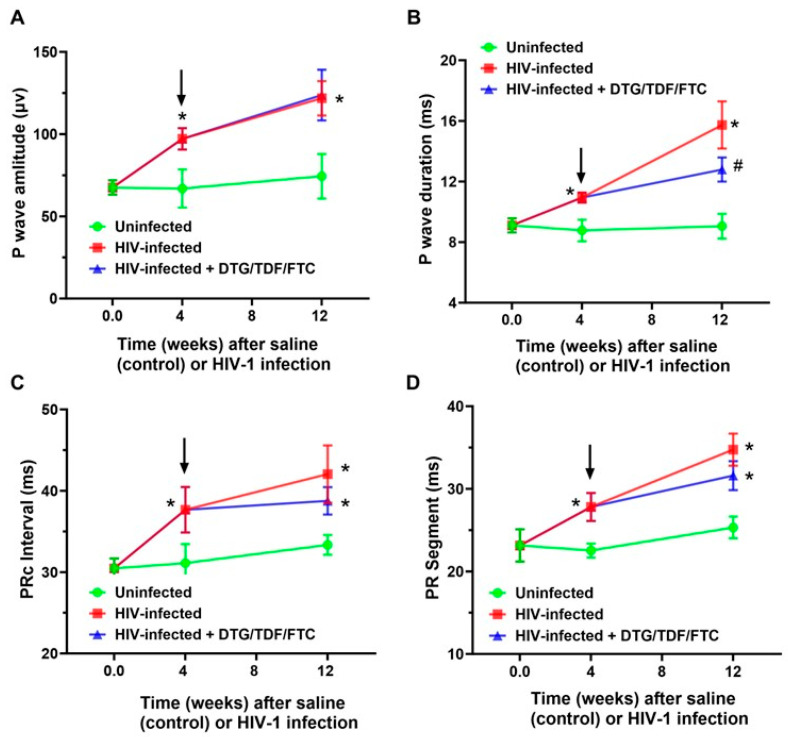
Changes in atrial electrical activity of the ECG in hearts from uninfected controls (green), HIV-1-infected (red), and HIV-1-infected Hu-mice treated with DTG/TDF/FTC. Panels (**A**,**B**) show P wave amplitude and duration. Panel (**C**) shows corrected PRc interval and Panel (**D**) shows PR segment. Data shown are mean ± SEM, with n = 8 mice per group (four males and four females). The FUJIFILM Visual Sonics offline software Vevo LAB 5.7.1 was used to obtain parameter values. Black arrows indicate the start of DTG/TDF/FTC treatment. * Denotes significantly different from uninfected (control), *p* < 0.05; # denotes significantly different from HIV-1-infected, *p* < 0.05.

**Figure 5 ijms-27-00519-f005:**
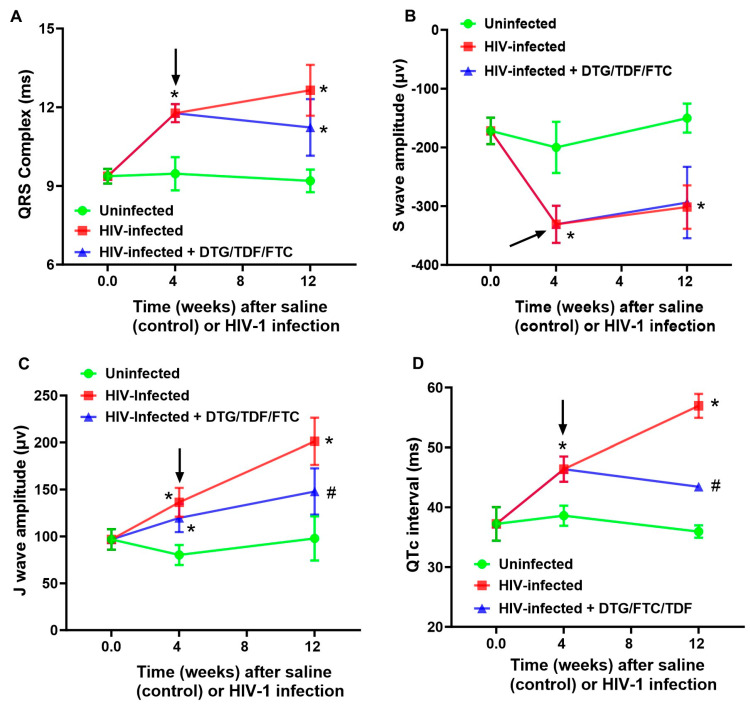
Changes in ventricular depolarization/repolarization activity of the ECG in hearts from uninfected controls (green), HIV-1-infected (red), and HIV-1-infected Hu-mice treated with DTG/TDF/FTC. Panel (**A**) shows QRS complex, Panel (**B**) shows the S wave amplitude, Panel (**C**) shows J wave amplitude and Panel (**D**) shows QTc interval. Data shown are mean ± SEM, with n = 8 mice per group, four males and four females. Black arrows indicate the start of DTG/TDF/FTC treatment. The FUJIFILM Visual Sonics offline software Vevo LAB 5.7.1 was used to obtain parameter values. * Denotes significantly different from uninfected (control), *p* < 0.05; # denotes significantly different from HIV-1-infected, *p* < 0.05.

**Figure 6 ijms-27-00519-f006:**
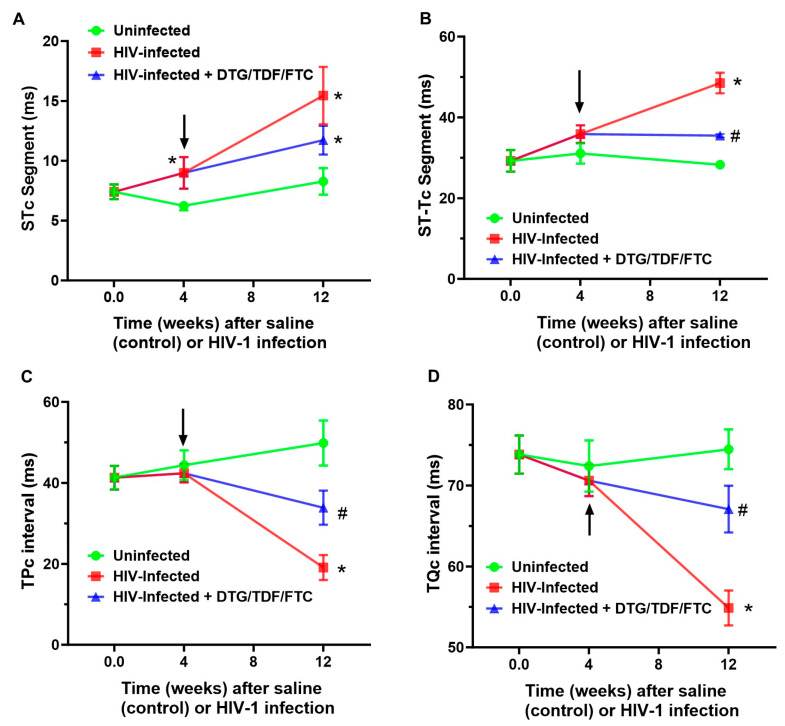
Changes in ventricular repolarization/electrical silent region of the ECG (latter) in hearts from uninfected controls (green), HIV-1-infected (red), and HIV-1-infected Hu-mice treated with DTG/TDF/FTC (blue). Panel (**A**) shows STc interval, Panel (**B**) shows the ST-Tc segment, Panel (**C**) shows TPc interval and Panel (**D**) shows TQc interval. Data shown are mean ± SEM, with n = 8 mice per group, four males, and four females. Black arrows indicate the start of DTG/TDF/FTC treatment. The FUJIFILM Visual Sonics offline software Vevo LAB 5.7.1 was used to obtain parameter values. * Denotes significantly different from uninfected (control), *p* < 0.05; # denotes significantly different from HIV-1-infected, *p* < 0.05.

**Figure 7 ijms-27-00519-f007:**
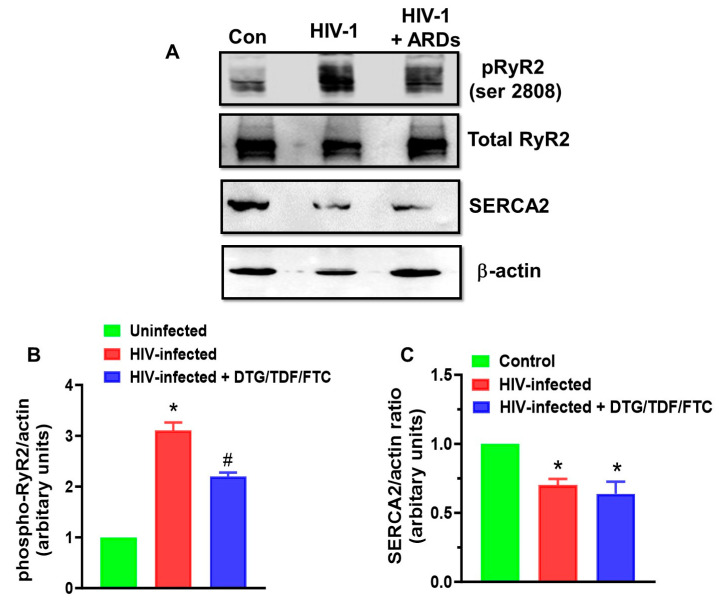
Panel (**A**) shows a representative autoradiogram for pRyR2 (Ser2808), total RyR2, SERCA2, and β-actin in ventricular tissues from uninfected controls, HIV-1-infected, and HIV-1-infected Hu-mice treated with DTG/TDF/FTC. Graph in Panel (**B**) is mean ± SEM, for pRyR2 (Ser2808) with n = 4–8 mice per group (males and females). Graph in Panel (**C**) is mean ± SEM, for SERCA2 with n = 4–8 mice per group (males and females). * Denotes significantly different from uninfected (control), *p* < 0.05; # denotes significantly different from HIV-1-infected, *p* < 0.05.

**Figure 8 ijms-27-00519-f008:**
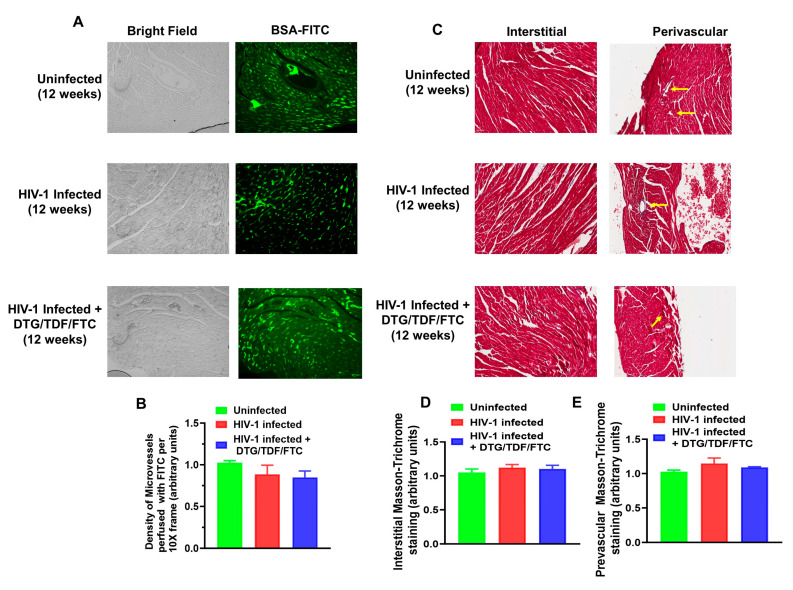
Panel (**A**) shows a representative 20× frame BSA-FITC (anterior apex) highlighting the density of perfused in ventricular tissues from uninfected controls, HIV-1-infected, and HIV-1-infected Hu-mice treated with DTG/TDF/FTC. Graph in Panel (**B**) is mean ± SEM, with n = 4–8 mice per group. Panel (**C**) shows representative Masson-trichrome staining (20×) used to assess fibrosis in ventricular tissues (interstitial and perivascular) from uninfected controls, HIV-1-infected, and HIV-1-infected Hu-mice treated with DTG/TDF/FTC. Graphs in Panels (**D**,**E**) are mean ± SEM, with n = 4–8 mice per group.

**Table 1 ijms-27-00519-t001:** Body weight (g) of animals used in the study during the 12-week protocol.

	Time = at Start	T= 4 Weeks	T = 8 Weeks	T = 12 Weeks
Uninfected Hu-mice (n = 8, 4 males and 4 females)	18.4 ± 1.1	18.9 ± 1.2	19.8 ± 1.2	19.6 ± 1.0
Hu-mice infected with HIV-1_ADA_ (n = 8, 4 males and 4 females)	19.8 ± 0.4	20.7 ± 0.4	21.7 ± 0.1	19.9 ± 0.7
Hu-mice infected with HIV-1_ADA_ + DTG/TDF/FTC (n = 8, 4 males and 4 females)	18.5 ± 1.4	19.7 ± 1.4	19.9 ± 1.4	19.1 ± 0.9

## Data Availability

The original contributions presented in this study are included in the article. Further inquiries can be directed to the corresponding author.

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
