# Peer review of "Prolonged QT Interval in HIV-1 Infected Humanized Mice Treated Chronically with Dolutegravir/Tenofovir Disoproxil Fumarate/Emtricitabine"

_ijms, 2026, doi:10.3390/ijms27010519_

Round 1

Reviewer 1 Report

Comments and Suggestions for Authors

Namvaran and colleagues show in the present manuscript that HIV infected humanized mice display changes to ECG parameters that have been implicated in sudden cardiac death (SCD). Those changes are found to be progressive during the 12 week period of the experiment, and can be in part prevented by initiating combination anti-retroviral treatment (cART) four weeks after HIV infection. The authors find that microvasculature and muscle tissue of the heart appear structurally unaffected but that muscular SERCA expression is diminished and RyR2 is more phosphorylated in both treated and untreated HIV infection than under control conditions. Those findings suggest an explanation for the extended QT intervals and support a role in SCD. The study is straight forward, well explained and presented. Cardiological complications and SCD are more frequent in PWH than uninfected individuals even in the era of cART. While there exist differences between the murine and human cardiovascular system, which the authors clearly acknowledge, the study presents a valuable experimental model to address an understudied medical need, in particular considering that the part of the population living with HIV is also aging. There are a few points that need the authors attention: Results shown in Fig.7 and 8 have been switched in the text. The legend for Figure 7 mentions a red arrow that is not visible. Line 349 ‘PLW’ should read ‘PWH’.

Comments on the Quality of English Language

There is only some minor language editing required. 

Author Response

Reviewer #1 Namvaran and colleagues show in the present manuscript that HIV infected humanized mice display changes to ECG parameters that have been implicated in sudden cardiac death (SCD). Those changes are found to be progressive during the 12-week period of the experiment and can be in part prevented by initiating combination anti-retroviral treatment (cART) four weeks after HIV infection. The authors find that microvasculature and muscle tissue of the heart appear structurally unaffected, but that muscular SERCA expression is diminished and RyR2 is more phosphorylated in both treated and untreated HIV infection than under control conditions. Those findings suggest an explanation for the extended QT intervals and support a role in SCD. Comment #1: The study is straight forward, well explained and presented. Response #1: Thank you. Comment #2: Cardiological complications and SCD are more frequent in PWH than uninfected individuals even in the era of cART. While there exist differences between the murine and human cardiovascular system, which the authors clearly acknowledge, the study presents a valuable experimental model to address an understudied medical need, in particular considering that the part of the population living with HIV is also aging. Response #2. We appreciate the comment by the reviewer and recognition of the gaps in models to delineate mechanisms that contribute to increased incidence of sudden cardiac death in PWH. Comment #3 There are a few points that need the authors attention: Results shown in Fig.7 and 8 have been switched in the text. The legend for Figure 7 mentions a red arrow that is not visible. Line 349 ‘PLW’ should read ‘PWH’. Response #3: Thank you for bringing this to our attention. All have been corrected. Thank you!

Reviewer 2 Report

Comments and Suggestions for Authors

Ali Namvaran and colleagues used HIV-infected humanized mice as a model and found that after DTG/TDF/FTC treatment, there was a prolongation of the QT interval, which was associated with increased RyR2 Ser2808 phosphorylation and decreased SERCA2 expression. This model can simulate electrocardiogram changes observed in people living with HIV and provides a tool for mechanistic studies. The overall quality of the manuscript is high and meets publication standards. However, the following issues must be addressed before publication:

  1. In this model, QT interval prolongation begins 4 weeks post-infection, which is earlier than that observed with drug treatment. Does the HIV-1 protein (such as Tat) directly regulate RyR2/SERCA2 expression at this stage? Therefore, it is necessary to supplement this study with viral protein intervention experiments to verify this.
  2. DTG/TDF/FTC treatment only prevents further worsening of the QT interval but does not restore it to baseline. Could this be due to the persistent suppression of SERCA2 expression by the drugs? It is necessary to add an “uninfected + drug” group to rule out the direct effects of the drugs.
  3. The incidence of SCD is higher in males with HIV (PWH); however, the study did not separately analyze differences between male and female mice in the QT interval and RyR2 phosphorylation levels. Is there a sex-specific mechanism? The background cites literature indicating that “the incidence of SCD in male HIV-infected patients (PWH) is 18% higher than in females,” but does not analyze the impact of sex on QT prolongation and calcium regulatory protein abnormalities using data from both male and female mice in this study.
  4. At 12 weeks post-infection, microvascular density and fibrosis in the myocardium showed no difference, but previous studies have shown related pathological changes emerging at 16–17 weeks post-infection. Does long-term infection (>12 weeks) exacerbate QT interval prolongation through ischemia or fibrosis? Therefore, it is necessary to extend the observation period.
  5. Increased RyR2 Ser2808 phosphorylation is associated with β-adrenergic activation. If β-blockers are used as an intervention, can QT interval prolongation be reversed? Functional verification experiments should be conducted.
  6. Aside from DTG/TDF/FTC, do other antiretroviral drugs (such as NNRTIs like efavirenz) also induce similar calcium regulatory abnormalities and QT interval prolongation? Can this model be used to screen for the cardiovascular safety of drugs?
  7. Western blot analysis used only three mice per group (instead of eight per group as in the overall experiment), which may result in insufficient representativeness of the findings for RyR2 phosphorylation and SERCA2 expression, and weak statistical reliability due to the small sample size.
  8. Only the expression/phosphorylation status of RyR2/SERCA2 was assessed; functional indicators such as Ca²+leakage during diastole and SERCA2 calcium pump activity in cardiomyocytes were not directly measured, making it impossible to directly link molecular changes to the QT interval prolongation phenotype.
  9. The background section requires revision. The authors should clarify the global use of DTG/TDF/FTC as a WHO first-line regimen and the clinical incidence of QT interval prolongation associated with this combination in PWH, thereby highlighting the study’s clinical significance. The current background mentions K⁺ current abnormalities and inflammation as relevant to QT prolongation but does not emphasize the research gap regarding “calcium regulatory protein (RyR2/SERCA2) abnormalities” in HIV-related QT prolongation. Prior studies in this field should be supplemented to define the innovations of this study.
  10. The discussion section should be expanded. The current discussion only broadly mentions “controlling viral load” and should specifically address the link between RyR2 phosphorylation and β-adrenergic activity, proposing the clinical hypothesis that “for PWH with QT prolongation, β-blockers may be considered.” Regarding “mouse-human electrophysiological differences,” the authors could suggest future studies using human induced pluripotent stem cell–derived cardiomyocytes (hiPSC-CMs) to validate the mechanisms; regarding the “impact of anesthesia,” it should be stated that awake mouse telemetry will be adopted in future experiments for an optimized study design.

Author Response

Response to Reviewer #2 Comments

Ali Namvaran and colleagues used HIV-infected humanized mice as a model and found that after DTG/TDF/FTC treatment, there was a prolongation of the QT interval, which was associated with increased RyR2 Ser2808 phosphorylation and decreased SERCA2 expression.

Comment #1: This model can simulate electrocardiogram changes observed in people living with HIV and provides a tool for mechanistic studies.

Response #1: We appreciate your recognition of the value of our work.

Comment #2: The overall quality of the manuscript is high and meets publication standards.

Response #2: We appreciate the positive comment.

Comment #3: In this model, QT interval prolongation begins 4 weeks post-infection, which is earlier than that observed with drug treatment.

  • Does the HIV-1 protein (such as Tat) directly regulate RyR2/SERCA2 expression at this stage?
  • Therefore, it is necessary to supplement this study with viral protein intervention experiments to verify this.

Response # 3.1.

Thank you for the outstanding questions. We do not know if HIV-1 proteins (such as Tat) can regulate the expression of RyR2/SERCA2. Determining this will require dedicated in vitro experiments using multiple cardiac myocyte cell lines and iPSC-derived cardiomyocytes, which we plan to pursue in a future study. It is important to note that such experiments must be carefully designed, given the relatively slow turnover rates of RyR2 (~9 days) and SERCA2 (~14 days) (Ferrington et al., J Biol Chem 273:5885–5891).

In this in vivo study we did not observe any changes in expression of RyR2 up to 12 weeks of infection without and with antiretroviral treatment, therefore, it is unlikely that the expression of RyR2 is regulated by HIV-1 proteins, including HIV-Tat. Nonetheless, we will conduct these experiments in separate studies to affirm that the expression of RyR2 in our model is not biphasic.

In an earlier study (Int. J. Mol. Sci. 2023, 24, 274) we showed that HIV-1 Tat can elicit spontaneous Ca2+ transients in isolated primary rat cardiac myocytes. We also showed that that HIV-Tat increased the open probability of RyR2 by increasing the dwell time in the open state (using lipid bilayer assays,). We have not evaluated whether other HIV-1 proteins including gp120 and Nef will also perturb intracellular Ca2+ homeostasis and modulate the activity of RyR2. These will be assessed in future in vitro studies using multiple cardiac myocyte cell lines and induced pluripotent stem (iPSCs)-derived cardiac myocytes.

At this time, we also do not know whether HIV-1 proteins (such as Tat) decrease SERCA2 expression. As stated in the Discussion, SERCA2 is known to be regulated by thyroid hormones (T3/T4) and is affected in hypothyroidism (Clin Infect Dis 2003, 37(4):579–583), as well as by the HIF-1α/miRNA-29c axis (Am J Physiol Heart Circ Physiol 2019, 316(3):H554–H565). These regulatory mechanisms will be investigated further in a follow-up study. In addition, whether HIV-Tat influences T3/T4 signaling or HIF-1α will be assessed in vitro using multiple cardiac myocyte cell lines and iPSC-derived cardiomyocytes.

Response # 3.2:   We agree that supplementing the study with viral protein intervention would provide additional insight. However, this work represents a first-of-its-kind in vivo study, and the new findings naturally raise further questions that extend beyond the scope of a single manuscript for IJMS. These follow-up experiments will be incorporated into future research proposals and subsequent studies.

This is also a first-of-its-kind in vivo study and new data raises new questions, not all of which could be included in a single manuscript for IJMS.

Comment 4: DTG/TDF/FTC treatment only prevents further worsening of the QT interval but does not restore it to baseline.

  • Could this be due to the persistent suppression of SERCA2 expression by the drugs?
  • (2) It is necessary to add an “uninfected + drug” group to rule out the direct effects of the drugs.

Response # 4.1: Thank you for pointing this out. Thus far, we have not investigated whether the antiretroviral drugs are down regulating SERCA2 expression. However, we suspect DTG/TDF/FTC combination and/or select agent from this combination could potentially have a direct effect. Recently we showed that DTG, TDF, and FTC upregulates HIF-1a in an oxygen-independent manner (Int. J. Mol. Sci. 202526(8), 3801).

Increased HIF-1α can elevate miRNA-29c levels, which in turn can downregulate SERCA2 expression (Am J Physiol Heart Circ Physiol 2019, 316, (3), H554-H565). We plan to validate these mechanistic links in future studies.

Response # 4.2:  Your suggestion to include uninfected + drug group to rule out the direct effects of the drugs is noted. New and exciting data raises more questions than answers, all of which cannot be addressed in one manuscript. When planning the study our initial objective was to determine if HIV-1-infected NOD.Cg-PrkdcscidIl2rgtm1Wjl/SzJ humanized mice (Hu-mice) treated with the first-line antiretroviral regimen, dolutegravir (DTG)/tenofovir disoproxil fumarate (TDF)/emtricitabine (FTC) can recapitulate the prolongation of the QT interval reported in PWH, Therefore, our comparison was intentionally limited to infected untreated versus infected + ART, mirroring the clinical scenario.

During cardiac biochemical analysis, we observed downregulation of SERCA2. In HIV-1–infected mice, we suspect this may be mediated by a hypothyroid-like mechanism (Clin Infect Dis 2003, 37(4):579–583). Given that DTG, TDF, and FTC also upregulate HIF-1α (Int. J. Mol. Sci. 2025, 26(8):3801), an alternate mechanism involving HIF-1α/miRNA-29c may be operative in ART-treated animals. These possibilities will be evaluated in follow-up in vitro studies using multiple cardiac myocyte cell lines and iPSC-derived cardiomyocytes. Such experiments must be carefully planned due to the slow turnover rate of SERCA2 (>10 days) (Ferrington et al., J Biol Chem 1998, 273:5885–5891).

Comment # 5: The incidence of SCD is higher in males with HIV (PWH); however, the study did not separately analyze differences between male and female mice in the QT interval and RyR2 phosphorylation levels.

(1) Is there a sex-specific mechanism?

(2) The background cites literature indicating that “the incidence of SCD in male HIV-infected patients (PWH) is 18% higher than in females,” but does not analyze the impact of sex on QT prolongation and calcium regulatory protein abnormalities using data from both male and female mice in this study.

Response #5.1: Thank you for raising this point. We have revised the text in the introduction as it is not clear at this time if there is a sex-specific mechanism. Data published online in Heart News: September 08, 2021 (American Heart Association) indicate an increase prevalence of SCD in males infected with HIV-1 compared to females. Whether the increase incidence in males is due to an increase in behavioral risks, an  increased health risks or more males than females were in their study  remains to be defined. In another Reinsch et al.,( HIV Clin Trials 2009;10(4):261–268) suggested that there might be a sex difference with women having slightly longer QT interval. In this study, we did not see any significant changes in ECG changes were not significantly different in this small group of males and females.

Response #5.2: Thank you for raising this point. Our data suggest that both males and females have reduced expression of SERCA2 and increased phosphorylation of RyR2 at Ser2808.

Comment #6: At 12 weeks post-infection, microvascular density and fibrosis in the myocardium showed no difference, but previous studies have shown related pathological changes emerging at 16–17 weeks post-infection.

(1) Does long-term infection (>12 weeks) exacerbate QT interval prolongation through ischemia or fibrosis? Therefore, it is necessary to extend the observation period.

Response #6: We  have not assessed QT interval in HIV-1 infected mice with and without antiretroviral treatment 16–17 weeks post-infection. However, we suspect that ischemia and fibrosis would worsen exacerbate the prolongation in QT interval (Narla, Clin Cardiol 2021, 44, (3), 316-321 and Brouillette, et al., Can J Cardiol 2019, 35, (3), 310-319).

Comment #7: Increased RyR2 Ser2808 phosphorylation is associated with β-adrenergic activation.

  • If β-blockers are used as an intervention, can QT interval prolongation be reversed?
  • Functional verification experiments should be conducted.

Response #7.1: Great question. This is what we suspect since RyR2 phosphorylation at Ser2808 is associated with β-adrenergic activation. As such, β-blocker intervention would potentially mitigate QT interval prolongation by reducing PKA-mediated phosphorylation of RyR2 at Ser2808. Alvi et al., (JACC Heart Fail 2019, 7, (9), 759-767). reported earlier that in older people with heart failure and HIV (PWH), b-blockers attenuated the hyperadrenergic function to decrease the risk of sudden cardiac death. Now that we have a model, we could not only validate these findings but will also help us to define the mechanism other mechanisms that contribute to the increased risk of sudden cardiac death in PWH.

Response #7.2:  Agree. We have started these experiments by implanting telemetric electrodes to assess ECG recordings in freely moving HIV-1 infected humanized mice to avoid the effects of anesthesia. Our plan was to treat infected animals with a β-blocker (Metoprolol) and/or with carvedilol (a nonselective beta-blocker with alpha-1 blocking properties) to determine and determine if these drugs would blunt ventricular tachyarrhythmias. While the telemetric electrodes (Data Science International) were accommodated subcutaneously in NSG mice (parent strain), they we too large for NSG humanized mice. As such we are awaiting approval IACUC to abdominally implant DSI telemetric ECG electrodes in our humanized mice. It is important to mention that now that a model is available, in the future we can proceed to address many unanswered questions in the area of HIV and sudden cardiac death.

Comment #8:

(1) Aside from DTG/TDF/FTC, do other antiretroviral drugs (such as NNRTIs like efavirenz) also induce similar calcium regulatory abnormalities and QT interval prolongation?

(2) Can this model be used to screen for the cardiovascular safety of drugs?

Response #8.1:  This is an excellent question, and we suspect efavirenz does. In a recent publication we showed that efavirenz elicited spontaneous Ca2+ transients in isolated primary rat cardiac myocytes (Int. J. Mol. Sci. 2023, 24, 274). In that study we also showed that efavirenz also binds to and increase and then decreased the open probability of RyR2. When HIV-Tat is present, the ability of  efavirenz to close RyR2 is enhanced. So yes, we suspect that efavirenz can increase the QT interval by inducing a gain-of-function of RyR2. Efavirenz (and other NNRTIs), can also indirectly impair sarcoplasmic reticulum Ca2+ cycling by impairing mitochondrial function and inducing oxidative stress. Our model can also be used to assess ECG changes with NNRTI as well as other classes of antiretroviral drugs and delineate mechanisms by which they do so.

Response $8.2  Yes this model can be used to screen for cardiovascular safety of drugs.

Comment #9: Western blot analysis used only three mice per group (instead of eight per group as in the overall experiment), which may result in insufficient representativeness of the findings for RyR2 phosphorylation and SERCA2 expression, and weak statistical reliability due to the small sample size.

Response #9:  We apologize for not fully explaining this. Multiple Western blots were conducted and the data shown was a representative. All raw Western blots were supplied to IJMS for verification of data integrity, and they were approved by Elena Peng from IJMS. The data shown in previous Figure 8 (now Figure 7) are representative RyR2, pRyR2 (Ser2808) and SERCA2 in ventricular homogenates from a control, HIV-1 infected and an HIV-1 infected humanized mice treated with DTG/TDF/FT. Below is the raw data for the multiple blots conducted.

Images showing raw for auto autoradiogram used to construct new Figure 7

Comment #10: Only the expression/phosphorylation status of RyR2/SERCA2 was assessed; functional indicators such as Ca²+leakage during diastole and SERCA2 calcium pump activity in cardiomyocytes were not directly measured, making it impossible to link molecular changes to the QT interval prolongation phenotype.

Response #10: This is an excellent question. When we were designing and planning the study, the scientific question being asked was whether HIV-1-infected NOD.Cg-PrkdcscidIl2rgtm1Wjl/SzJ humanized mice (Hu-mice) treated with the first-line antiretroviral regimen, dolutegravir (DTG)/tenofovir disoproxil fumarate (TDF)/emtricitabine (FTC) can recapitulate the prolongation of the QT interval reported in PWH. After longitudinally assessing ECG, hearts were from excised from animals cut in halves and one half was placed in paraformaldehyde and the other half was flash frozen and stored at -80oC for biochemical assays. The Western blots shown below were done with flash frozen tissues. Now that we have data showing altered expression/phosphorylation state of SERCA2 and RyR2, we will begin to assess Ca2+ leak using freshly isolated myocytes for which we have the expertise to do (Shao et al., J Mol Cell Cardiol2007 Jan;42(1):234-46.Shao et al., 2012, 82(3) p383-399).  These data will be in another manuscript.

It should also be mentioned that our UNMC’s Biosafety Committee are limiting our ability to isolate cardiac myocytes from HIV-infected animals for assessing Ca2+ transients using traditional continuous Langendorff perfusion method in a P3 biosafety room. Currently we are optimizing methods to isolate myocytes using collagenase in a dish.

Several laboratories have already demonstrated a link between gain-of-function of RyR2 arising from increased phosphorylation and prolongation in the QT interval (cited in references 29, 30, 31, 32, 33, 34, 35, 36, 56). A reduction in expression of SERCA2 will pump activity and delay myocyte relaxation, as cited in the discussion (references 66 and 67).

Comment #11: The background section requires revision.

(1) The authors should clarify the global use of DTG/TDF/FTC as a WHO first-line regimen and the clinical incidence of QT interval prolongation associated with this combination in PWH, thereby highlighting the study’s clinical significance.

(2) The current background mentions K⁺ current abnormalities and inflammation as relevant to QT prolongation but does not emphasize the research gap regarding “calcium regulatory protein (RyR2/SERCA2) abnormalities” in HIV-related QT prolongation.

(3) Prior studies in this field should be supplemented to define the innovations of this study.

Response #11.1: Agree! In July 2019, the WHO recommended dolutegravir as preferred HIV treatment option in all populations (https://www.who.int/news/item/22-07-2019-who-recommends-dolutegravir-as-preferred-hiv-treatment-option-in-all-populations), In December 2019,Clinicalinfor.gov (https://clinicalinfo.hiv.gov/en/guidelines/adult-and-adolescent-arv/what-start-initial-combination-regimens-antiretroviral-naive-1) and the British HIV Association (https://www.aidsmap.com/about-hiv/recommended-treatments-hiv) recommended an integrase strand inhibitor with two nucleoside reverse transcriptase inhibitors including dolutegravir plus tenofovir disoproxil fumarate and emtricitabine as a first line recommended regimen for antiretroviral-naïve patients.  The reason for using this drug combination and not “older drug combinations” is to assess future sudden cardiac death incidence.

The data from this study show that HIV-1 infection is what is driving prolongation in the QT interval and DTG/TDF/FTC treatment did not exacerbate the increase in QT interval. Our data agree with clinical studies showing that even supratherapeutic dose of dolutegravir (DTG), tenofovir disoproxil fumarate (TDF) and emtricitabine are not considered a QT-prolonging drug. (Chen et al., Pharmacotherapy 2012 Apr;32(4):333-9; Mu et al., Front Pharmacol 2023 Oct 31;14:1268597;  Florentini, et al., BMC Cardiovasc Disord 2012 Dec 23;12:124). It should mentioned that there are some studies that suggest (FTC) although FTC is not statistically associated with QTc prolongation  Whether FTC in combination with other antiretrovirals like tenofovir is associated with QTc prolongation remains to be determined. 

The introduction has been revised to reflect these changes.

Response #11.2. We have revised the introduction section to better emphasize while the emphasis on increased incidence on sudden cardiac death has been linked to abnormalities in delayed rectifying and repolarizing K⁺ current, inflammation and fibrosis, little is known about whether changes in expression and function of sarcoplasmic reticulum Ca2+ proteins, namely RyR2 and SERCA2, also contribute to QT interval prolongation.

Response #11.3. The introduction has been revised to clearly define the innovation of this study.

Comment #12: (1) The discussion section should be expanded.

  • The current discussion only broadly mentions “controlling viral load” and should specifically address the link between RyR2 phosphorylation and β-adrenergic activity, proposing the clinical hypothesis that “for PWH with QT prolongation, β-blockers may be considered.”
  • Regarding “mouse-human electrophysiological differences,” the authors could suggest future studies using human induced pluripotent stem cell–derived cardiomyocytes (hiPSC-CMs) to validate the mechanisms; regarding the “impact of anesthesia,” it should be stated that awake mouse telemetry will be adopted in future experiments for an optimized study design.

Response #12.1: We thank the reviewer for the suggestion. We have included in the revised discussion that “Since there was an increase in RyR2 are Ser2808 and this residue on RyR2 is phosphorylated by protein kinase A-dependent activation of b-adrenergic receptors, we posit that β-blockers may be considered as an adjunct treatment for PWH to attenuate the gain-of-function of RyR2 and increased risk for ventricular tachyarrhythmia.

Response #12.2 Thank you. We have indicated in the revised discussion that in future studies using human induced pluripotent stem cell–derived cardiomyocytes (hiPSC-CMs) to validate the mechanisms. We have also indicated to minimize the impact of isoflurane anesthesia, animals will be implanted with telemetric electrode, so as to be able to assess ECG in freely moving conscious animals. Also see Response #7.2

Thank you!

Reviewer 3 Report

Comments and Suggestions for Authors

This manuscript aims to model QT interval prolongation observed in people with HIV using humanised mice under antiretroviral treatment. However, the study appears to have been prepared with limited scientific guidance and lacks the rigour expected for publication in this journal. The writing quality is insufficient for clear scientific communication, the structure is disorganised, and the overall presentation does not meet expected standards. Figures and tables are inconsistently formatted, legends are incomplete or missing, and several visual elements do not contribute meaningfully to the results. The study also remains largely descriptive, without providing new mechanistic or conceptual insights into QT interval abnormalities in HIV infection. Taken together, the manuscript does not demonstrate the level of novelty, clarity, or scientific quality required for publication in the journal.

Author Response

Reviewer # 3

This manuscript aims to model QT interval prolongation observed in people with HIV using humanized mice under antiretroviral treatment.

Comment #1: The study appears to have been prepared with limited scientific guidance and lacks the rigour expected for publication in this journal.

Response #1:  We respectively disagree. Our proof-of-concept rigorous study design and findings are of significance to the journal as clarified below.

  • The scientific question being asked in this study is whether HIV-1-infected NOD.Cg-PrkdcscidIl2rgtm1Wjl/SzJ humanized mice (Hu-mice) treated with the first-line antiretroviral regimen, dolutegravir (DTG)/tenofovir disoproxil fumarate (TDF)/emtricitabine (FTC) can recapitulate the prolongation of the QT interval reported in PWH.
  • To our knowledge, this is the first in vivo study to address this specific question, and we used eight mice per group (four males and four females), consistent with standards commonly applied in humanized mouse research. Although sex differences in QT interval have been variably reported in the literature (Heart News, American Heart Association, Sept 8, 2021; Reinsch et al., HIV Clin Trials 2009;10(4):261–268), we did not observe sex-based differences in ECG parameters in our dataset. For statistical power and clarity, the study groups were therefore analyzed without additional sex stratification.
  • Our results demonstrate, for the first time, that (a) HIV-1 infection alone prolongs the QT interval, and (b) HIV-1–infected Hu-mice treated with DTG/TDF/FTC also show QT-interval prolongation, mirroring findings in PWH. Additional ECG alterations observed in our study are likewise consistent with clinical reports. These findings advance understanding in an area where clinically relevant mechanistic models remain limited.

While the reviewer described the data as “descriptive,” humanized mouse studies inherently present unique technical and biosafety challenges, including the need for BSL-2+ facilities and the fragility of the model. The study was therefore designed with careful consideration to feasibility and rigor within these constraints. Importantly, all raw Western blot data were submitted and verified by the journal, supporting the integrity of the presented results.

We believe the novelty of the findings, the use of a clinically relevant model, and the alignment with reported ECG abnormalities in PWH collectively support the appropriateness of this work for publication in IJMS.

Comment #2: The writing quality is insufficient for clear scientific communication, the structure is disorganized, and the overall presentation does not meet expected standards.

Response #2:  The manuscript was reviewed by all authors and professionally edited prior to submission. While minor syntax or grammar issues may have been missed, these can be addressed during the revision process. Differences in regional English spelling may also contribute to perceived inconsistencies.

Regarding manuscript structure, we organized the study logically: ECG recordings, followed by microvascular perfusion, expression of RyR2, phospho-RyR2 (Ser2808), and SERCA2. and finally fibrosis. We believe this sequence appropriately reflects the flow of the experimental results.

To further improve clarity, we would welcome specific guidance from the reviewer regarding any structural changes or areas of confusion, rather than general comments, so that we can ensure the manuscript meets the standards expected by IJMS.

Comment #3: Figures and tables are inconsistently formatted, legends are incomplete or missing, and several visual elements do not contribute meaningfully to the results.

Response #3:  There is one table in the manuscript formatted according to the IJMS template. One figure originally had three panels in which the labels A, B, and C were inadvertently omitted; these have now been added. The other figures have an even number of panels and are grouped according to their respective ECG locations. The current formatting also accounts for IJMS’s print layout, which efficiently manages white space.

Comment #4: The study also remains largely descriptive, without providing new mechanistic or conceptual insights into QT interval abnormalities in HIV infection.

Response #4: We respectfully disagree with the characterization that the study is “largely descriptive.” While mechanistic insights are valuable, not all manuscripts need to be fully mechanistic to make a meaningful contribution. Many studies in the literature are hypothesis-generating, providing critical new observations that guide future research.

Our study provides several novel and important findings: for the first time, we demonstrate that in addition to previously reported contributions from Na⁺ and K⁺ channels, fibrosis, inflammation, and older antiretroviral drugs, HIV-1 infection and first-line ART induce by increasing phosphorylation of RyR2 at Ser2808, are likely inducing sympathetic-excitation, β-adrenergic receptor activation, and protein kinase A (PKA) activation.. It also shows for the first time downregulation of SERCA2. These findings advance our understanding of QT-interval abnormalities in HIV and provide a solid foundation for future mechanistic studies.

Given the inherent challenges of working with HIV-1–infected humanized mice and the novelty of these in vivo observations, we believe the study represents a significant contribution and is suitable for publication in IJMS.

Comment #5: Taken together, the manuscript does not demonstrate the level of novelty, clarity, or scientific quality required for publication in the journal.

Response #5: We respectfully disagree with the reviewer’s assessment that the manuscript lacks novelty, clarity, or scientific quality. The study provides several first-in-class findings, including demonstration that HIV-1 infection and first-line ART induce QT-interval prolongation, increased phosphorylation of RyR2 at Ser2808 (a PKA site) and downregulation of SERCA2.These new data suggest that starting b-blockers early may mitigate ventricular tachyarrhythmia by preventing gain-of-function of RyR2, provide a strong foundation for future mechanistic studies in humanized mice.

Without specific details on which aspects of novelty, clarity, or scientific quality are considered insufficient, the reviewer’s comment is difficult to address constructively. We believe the manuscript makes a significant contribution to the field and is appropriate for publication in IJMS.

Thank you!

Round 2

Reviewer 2 Report

Comments and Suggestions for Authors

The author has resolved my confusion and suggested publication.

Reviewer 3 Report

Comments and Suggestions for Authors

See comments to the Editor.